

# An expert study on hierarchy comparison methods applied to biological taxonomies curation

Lilliana Sancho-Chavarria[1], Fabian Beck[2] and Erick Mata-Montero[1]

[1] School of Computing, Costa Rica Institute of Technology, Cartago, Cartago, Costa Rica
[2] paluno, University of Duisburg-Essen, Essen, North Rhine-Westphalia, Germany

## ABSTRACT

Comparison of hierarchies aims at identifying differences and similarities between two or more hierarchical structures. In the biological taxonomy domain, comparison is indispensable for the reconciliation of alternative versions of a taxonomic classification. Biological taxonomies are knowledge structures that may include large amounts of nodes (taxa), which are typically maintained manually. We present the results of a user study with taxonomy experts that evaluates four well-known methods for the comparison of two hierarchies, namely, *edge drawing*, *matrix representation*, *animation* and *agglomeration*. Each of these methods is evaluated with respect to seven typical biological taxonomy curation tasks. To this end, we designed an interactive software environment through which expert taxonomists performed exercises representative of the considered tasks. We evaluated participants' effectiveness and level of satisfaction from both quantitative and qualitative perspectives. Overall quantitative results evidence that participants were less effective with *agglomeration* whereas they were more satisfied with *edge drawing*. Qualitative findings reveal a greater preference among participants for the *edge drawing* method. In addition, from the qualitative analysis, we obtained insights that contribute to explain the differences between the methods and provide directions for future research.

## INTRODUCTION

Visual comparison of hierarchies has been prevalent in information visualization research because it is relevant for a wide range of domains such as tracking changes in software projects, comparing budgets, and describing dynamics of organizational structures, among others. In this work, we study the comparison of hierarchies in the domain of biological taxonomies. Taxonomic information has been scattered in publications for centuries. In spite of integration efforts of global initiatives in the last decades, there are numerous taxonomic databases around the globe (*Ball-Damerow et al., 2019*). A visualization tool for the identification of differences and similarities between two versions of a taxonomy would contribute to such integration efforts.

Taxonomies are hierarchies created by experts to classify living organisms. Through classification, mutually resembling organisms are placed together in categories known as

Corresponding author
Lilliana Sancho-Chavarria,
lsancho@tec.ac.cr

*taxonomic ranks*, which, in turn, make up the levels of the hierarchy. The main taxonomic ranks include *domain*, *kingdom*, *phylum*, *class*, *order*, *family*, *genus* and *species*. Each node within the hierarchy is referred to as a taxon, that is, a name given to a group of organisms; for instance, *Vertebrates* and *Mammals* are two taxa, the former is placed at the *phylum* taxonomic rank and latter at the *class* taxonomic rank. Species and sub-species are placed at the lower level of the hierarchy and their scientific names are expressed with a *binomial system of nomenclature* that uses a Latin grammatical form. The first part is the genus and the second part is the *specific epithet*. For instance, the domestic cat's scientific name is *Feliz catus* (formerly known, for many years, as *Felix domesticus*), where the epithet name is *catus* and the genus is *Feliz*. Upper levels of the cat's taxonomy are: family *Felidae*, order *Carnivora*, class *Mammalia*, phylum *Chordata* and kingdom *Animalia*. For a taxon to be recorded in a taxonomy, it must have been described in a publication, either as a new group or as a review of an existing group of organisms. Taxonomy records may include various data; but for comparison, they are required to include at least taxon name, author's name, and the year of publication. This allows users to determine under which judgment the classification was devised. After almost three centuries since modern taxonomy was first established by *Von Carl & Friedrich (1767)*, one might think that most organisms on Earth have been identified and classified, and that taxonomies are rather static. However, on one hand, it is estimated that only about 1.5 million from approximately 11 million species of macro organisms haven been identified and described (*Larsen et al., 2017*). On the other hand, the dynamics of taxonomic work has lead experts worldwide to end up with different versions of the classifications. Taxa names represent concepts whose definition depends on the authors' criteria, which eventually gives rise to conflicting versions of a taxonomy. These multiple versions will require corrections and re-classifications in order to come to an integrated version that can more accurately document biodiversity. That is how taxonomists often face the problem of reconciling different versions of a taxonomy. For such reconciliation efforts, biological taxonomists require to perform a series of curation tasks.

*Sancho-Chavarria et al. (2016, 2018)* characterized curation tasks that involve taxonomic changes when comparing two versions $T_1$ and $T_2$ of a taxonomy. Such characterization involved interviews to six experts from three different countries and followed a two-stage analysis. During the first stage, the authors reviewed literature and interviewed experts in order to obtain a list of preliminary tasks. In the second stage, the tasks were shared and discussed with the experts, in order to obtain a final list of ten tasks. Table 1 provides a description of those derived ten tasks organized into three categories, namely, *pattern identification*, *query* and *edition*. Some of these tasks are domain-independent, and are frequently mentioned in information visualization research (e.g., filtering, focus, retrieving details). Other are domain-specific to biological taxonomy curation work (e.g., tasks in the pattern identification category such as identify splits or identify merges).

- *Pattern identification* tasks include the identification of *congruent*, *merged*, *split*, *renamed*, *moved*, and *added/excluded* taxa, as well as the *overview* of changes and the

visualization of a *summary* of the resulting comparison. Congruence refers to same taxonomic concepts present in both versions of a taxonomy. A split occurs when taxonomists determine that a group previously considered a unit actually consists of several groups of species that should be described separately. Conversely, a merge happens when taxonomists decide that several independent taxa should be combined into the same group. A change of name (i.e., rename) is usually due to a typo that needs to be corrected. A taxon appears moved when it has been re-classified and placed in a different location within the other version of the taxonomy. An addition occurs when a new taxon is added to the taxonomy, either because it is a new discovery or because it had not been previously recorded in the database. Exclusions refer to taxa that are contained in version $T_1$ but that are missing in the alternative version $T_2$. It is important to note that, from a taxonomic point of view, once a species is discovered, it is kept in the taxonomy even if the species becomes extinct; however, in this work we consider exclusions because it is important that taxonomists know when records are missing in the database. The *overview changes* task refers to the possibility of globally overviewing all differences between two versions of a taxonomy. The *summarize* task consists of obtaining statistical information on changes.

- Curation tasks in the *Query* category enable users to obtain detailed information on taxa. The *retrieve details* task lets users obtain attributes of a taxon, for instance, the year of publication and the authors' names. The *focus* task refers to the action that users perform when focusing on a group of organisms. Through *filter*, users may find taxa that satisfy some given conditions and through *find inconsistencies*, users may recognize differences due to errors or missing information (e.g., typos or undefined names).

- The *Edition* category comprises just one task that is rather ample, namely, the process through which experts make changes to $T_1$ and/or $T_2$ after analyzing the results of a comparison.

For this research, we considered tasks that are most relevant for the identification of changes between two versions of a taxonomy, that is, (1) *identify congruence*, (2) *identify corrections* (splits, merges, moves and renames), (3) *identify additions/exclusions* and (4) *overview changes*. Changes are more likely to occur at lower level taxonomic ranks, for instance, species level (*Vaidya, Lepage & Guralnick, 2018*), therefore for this study the identification of changes will be visualized only at species level.

Previous research has contributed with visual models and tools to support the comparison of alternative versions of a taxonomy (*Contian et al., 2016*; *Graham, Craig & Kennedy, 2008*; *Dang et al., 2015*). However, in practice, most taxonomists still rely on simple indented lists to carry out the curation process with little computational assistance. Information visualization provides visual representations that help people perform their tasks more efficiently (*Munzner, 2015*); therefore, we believe that the identification of differences can be eased with the support of a hierarchy comparison visualization system.

*Graham & Kennedy (2010)* surveyed the comparison of hierarchies and organized the visualization methods into five categories, namely, *edge drawing*, *coloring*, *animation*,

| Table 1 Biological taxonomy curation tasks. | |
| --- | --- |
| **Category** | **Task** |
| Pattern identification | (1) Identify congruence: Identify same taxonomic concepts. |
| | (2) Identify corrections: Identify splits, merges, moves, renames. |
| | (3) Identify additions/exclusions: Identify new or missing taxa. |
| | (4) Overview changes: Obtain a global view of changes. |
| | (5) Summarize: Obtain numerical understanding of change. |
| Query | (6) Find inconsistencies: Recognize violation of rules (e.g., repeated names). |
| | (7) Filter: Find cases that satisfy certain conditions. |
| | (8) Retrieve details: Retrieve the attributes of a particular concept. |
| | (9) Focus: Navigate and see the information in detail. |
| Edit | (10) Edit: Make changes to the taxonomies. |

*matrix representation* and *agglomeration*. The *edge drawing* method presents the two hierarchies as separate structures where differences and similarities are represented by edges from nodes in $T_1$ to the associated nodes in $T_2$. The *coloring* method represents similar nodes with the same color. *Animation* shows changes as smooth transitions from one hierarchy to the other. In a *matrix* representation, one hierarchy is placed along the vertical axis and the other one along the horizontal axis; matrix cells indicate relationships between nodes of the compared hierarchies. The *agglomeration* method visually merges both hierarchies into an integrated list.

The open question that we address in this work is how well these methods support the above curation tasks between two versions $T_1$ and $T_2$ of a taxonomy. From the five methods, we leave out coloring as an independent condition since color can be used across all methods. We designed and conducted a user study where 12 expert taxonomists evaluate those four methods. We wanted participants to interact with each of the four methods in a close-to-reality scenario. We developed an interactive software environment that integrates the four methods and allows users to easily navigate from one method to another while doing the assessment exercises (see Figs. 1–4). We wanted to capture the essence of each method and avoided the introduction of features that could potentially favor any particular method. We developed the software taking into consideration the importance of reaching a balanced implementation; thus, the software included same functionality and a common user interface design for all methods. We also carefully selected the data. Datasets contain sufficient types of changes to carry out exercises for all tasks and were also selected to avoid the introduction of any bias due to the potential prior knowledge of the data by the experts. Participants performed the same exercises with each method; however, the target taxa were not the same, also to avoid bias from a learning effect. Immediately after performing the exercises related to a task, participants were asked to answer a user satisfaction questionnaire in order to evaluate each method in relation to the completion of that task. We registered the participants' answers along with their interactions and thinking out-loud comments. We performed a quantitative analysis on the participants' responses to the exercises (i.e., whether they answered

correctly or not) and also on their user satisfaction assessment. Additionally, we obtained qualitative findings based on the participants' feedback throughout the session.

Our contribution with this work is twofold. On one hand, we assessed the effectiveness and level of participants' satisfaction with each visualization method. Participants were likely effective with three methods: *edge drawing*, *matrix* and *animation*, and also preferred *edge drawing* over the other methods. On the other hand, we obtained a set of themes that contribute to explain the differences between the methods and provide valuable insights for future work on the design of software tools for the comparison of biological taxonomies.

The rest of this article is organized as follows. Section 2 presents "Related Work". Section 3 details the "Study Design". It includes a description of the interactive software environment, the participants' profile, the characteristics of the datasets, the user study protocol and the questionnaire. Section 4 describes the "Results" of the study. In Section 5 "Discussion", we discuss the results and present limitations and implications of the study. Finally, in Section 6, we present "Conclusions and Future Work".

## Related work

Comparison—understood as the examination of two or more items to determine similarities and differences—is a means that facilitates the process of interpreting information. *Gleicher et al. (2011)* provide a comprehensive survey of visual comparison approaches focusing on comparing complex objects. They analyze a number of publications, systems and designs, looking for common themes for comparison, and propose a *general categorization of visual designs for comparison* that consists of three general categories, namely, juxtaposition, superposition and explicit encoding. The juxtaposed designs place the objects to be compared separately, in time or space. The superposed designs place the objects to be compared one over the other in an overlay fashion. The explicit encoded designs show the relationships between objects explicitly and is generally used when the relationships between objects are the subject of comparison. Hybrid designs are also possible and usually combine two categories. More recently, a set of considerations to understand comparison tasks and their challenges has been discussed (*Gleicher, 2018*), as well as comparison in the context of exploratory analysis (*Von Landesberger, 2018*). *Guerra-Gómez et al. (2012)* provide a classification of trees for comparison. The trees used in this work are similar to Type 0 trees, which display only a label; however, nodes also contain information about the author and year of publication of the species. Each node also contains the list of synonyms. All these data are necessary in order to identify the types of changes.

As mentioned above, for the comparison of biological taxonomies we are considering the methods surveyed by *Graham & Kennedy (2010)* for the comparison of two hierarchies, namely, *edge drawing*, *matrix*, *animation* and *agglomeration*. Each of these methods can be mapped to the mentioned *general categorization of visual designs for comparison*. So, the *edge drawing* method comprises characteristics from both juxtaposition (hierarchies are placed separately side by side) and explicit encoding (edges encode the relations between nodes); the *matrix* layout corresponds to an explicit

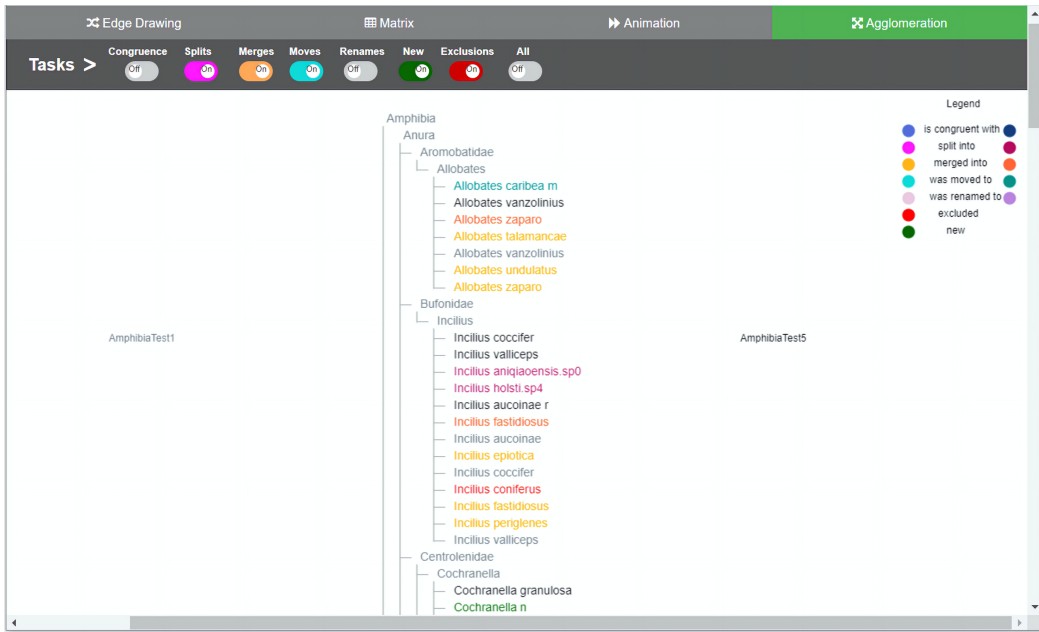

**Figure 1 Agglomeration method implementation.** Taxonomies are agglomerated into one hierarchical structure. Changes are indicated by relationships between the two taxonomies. In order to distinguish to which taxonomy the taxon belongs to, a dual use of color was introduced. For each taxon's specific change, light nuanced nodes indicate that the taxa belongs to the taxonomy of origin and the darker nuanced nodes indicate that they belong to the taxonomy of destination. Through the colored toggle buttons on the menu, users can visualize either one or several types of changes at once. Users can select a specific taxon in order to visualize its changes.           

encoding design since the matrix cells can explicitly indicate the relations among taxa; and *agglomeration* corresponds to a superposed design.

Previous works on hierarchy comparison match these categories. For instance, TreeJuxtaposer (*Munzner et al., 2003*) compares phylogenetic trees by using a juxtaposed layout. It presents a novel focus + context technique for guaranteed visibility and comparison is approached by coloring. *Holten & Van Wijk (2008)* present a visualization method where hierarchies are structured as icicle plots placed in juxtaposition. Relations are explicitly represented by edges arranged through hierarchical edge bundles to reduce cluttering. The Taxonomic Tree Tool (*Contian et al., 2016*) uses a juxtaposed layout to compare biological taxonomies. It combines glyphs to explain the relations between taxa. ProvenanceMatrix (*Dang et al., 2015*) compares two taxonomies using a matrix representation. Relations are explicitly displayed through two mechanisms: glyphs and edges. *Beck & Diehl (2010)* compare two software architectures that use a matrix; hierarchies here are represented as icicle plots.

Examples that use animation for comparison are scarcer. *Ghoniem & Fekete (2001)* use animation to visualize the transition between two alternative representations of the same tree laid out as treemaps. Considering agglomeration-based designs, *Beck et al. (2014)* present a nested icicle plot approach for comparing two hierarchies and *Guerra-Gómez et al. (2012)* contrast two trees for the visualization of both node value changes as well as topological differences for the comparison of budgets. *Lutz et al. (2014)* compared

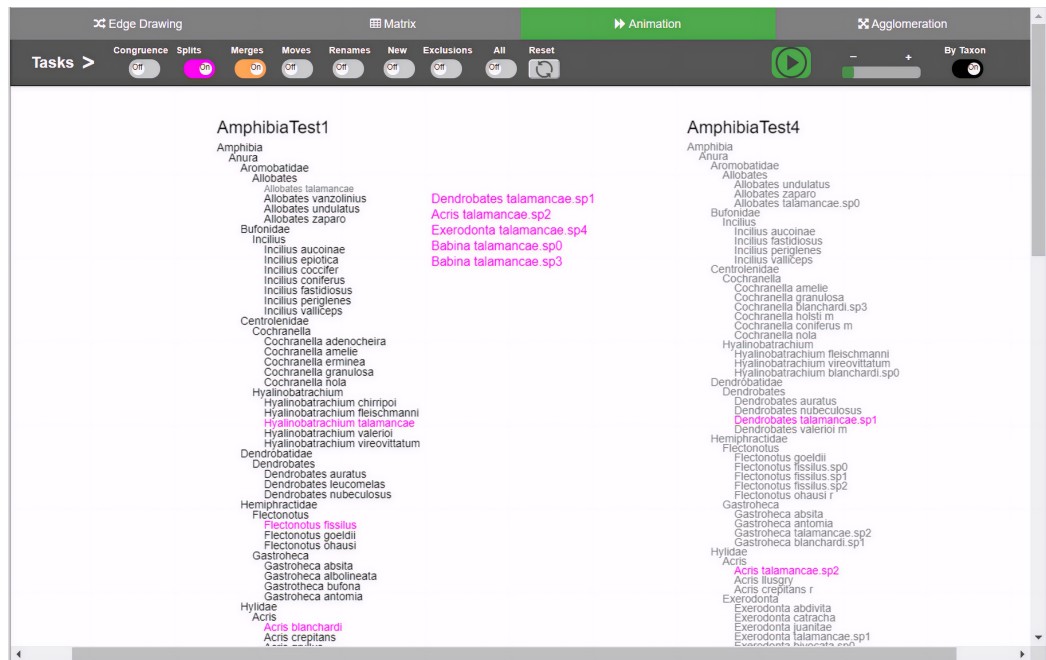

**Figure 2 Animation method implementation.** Changes are presented through animation. Besides the common toggle buttons on the main menu for all methods, the animation method includes a play/stop button, a speed slider, and a button that controls whether the animation will present the changes either one by oner or simultaneously. During animation, taxa moves from the origin taxonomy on the left side to the destination taxonomy on the right, displaying the transformation. Through the colored toggle buttons on the menu, users can animate either one or several types of changes at once. Users can select a specific taxon in order to visualize its changes.        

directory structures and conducted a qualitative user study to identify usage strategies. Unlike the above-mentioned work, which did not focus on biological taxonomies, *Graham & Kennedy (2007)* propose an agglomerated visualization based on directed acyclic graphs for the comparison of multiple biological taxonomies. They also analyzed set-based hierarchies and agglomerated graph-based visualizations for the comparison of botanical taxonomies (*Graham, Kennedy & Hand, 2000*).

Our work differs from the above-mentioned previous studies in that, for the first time, four visualization methods described in (*Graham & Kennedy, 2010*), are assessed for the comparison of pairs of biological hierarchies with respect to typical curation tasks.

## Study design

The research question addressed is: *How well does each method support carrying out biological taxonomy curation tasks?* This is assessed both quantitatively and qualitatively. On one hand, we aim at obtaining a quantitative understanding of the participants' effectiveness and level of satisfaction. On the other hand, we also aim at obtaining qualitative insights on the capacity of each method to carry out tasks for the comparison of biological taxonomies. We therefore explore how users interact with the visualizations and what their judgment of each method is. We opted for a within-subject design study that involves four experimental conditions, one for each method (*edge drawing*, *matrix*,

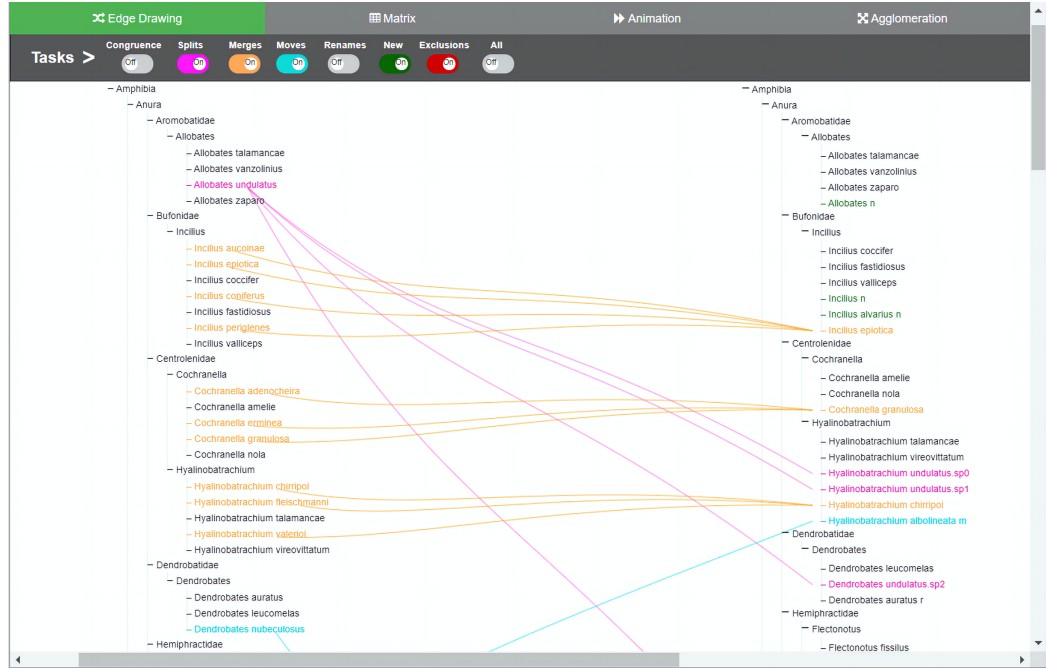

**Figure 3 Edge drawing method implementation.** Changes are indicated by relationships between the two taxonomies. In the edge drawing implementation, relationships are visualized by color-coded edges that go from one taxonomy to the other. Through the colored toggle buttons on the menu, users can visualize either one or several types of changes at once. Users can select a specific taxon in order to visualize its changes.

*animation* and *agglomeration*). In this way, each participant could test and contrast all methods. The study fits into the category "Evaluating Visual Data Analysis and Reasoning" (*Lam et al., 2012*). In this approach, the goal is to assess how a visualization tool supports the analytic process for a particular domain. Accordingly, in our study we want to evaluate how each implemented method supports the identification of similarities and differences for the curation of biological taxonomies. In order to assess each method for each task individually, participants were asked to solve a task with each method, then giving feedback, before moving to the next task. We favored this design over one where participants would first complete all tasks with one method (before moving to the next method) because we wanted to gain task-specific insights on the differences of the tested methods.

## The software environment

We developed an interactive web-based software environment that integrates in the same environment the four methods for visual comparison of biological taxonomies described in the Introduction section. The software environment was designed to investigate how these four methods support the taxonomy curation tasks described in Table 1. We tried to conceive a balanced design within the same environment, in order not to favor any method. We also decided to develop a functional system in which participants get a realistic impression. We provided remote web-based access to the

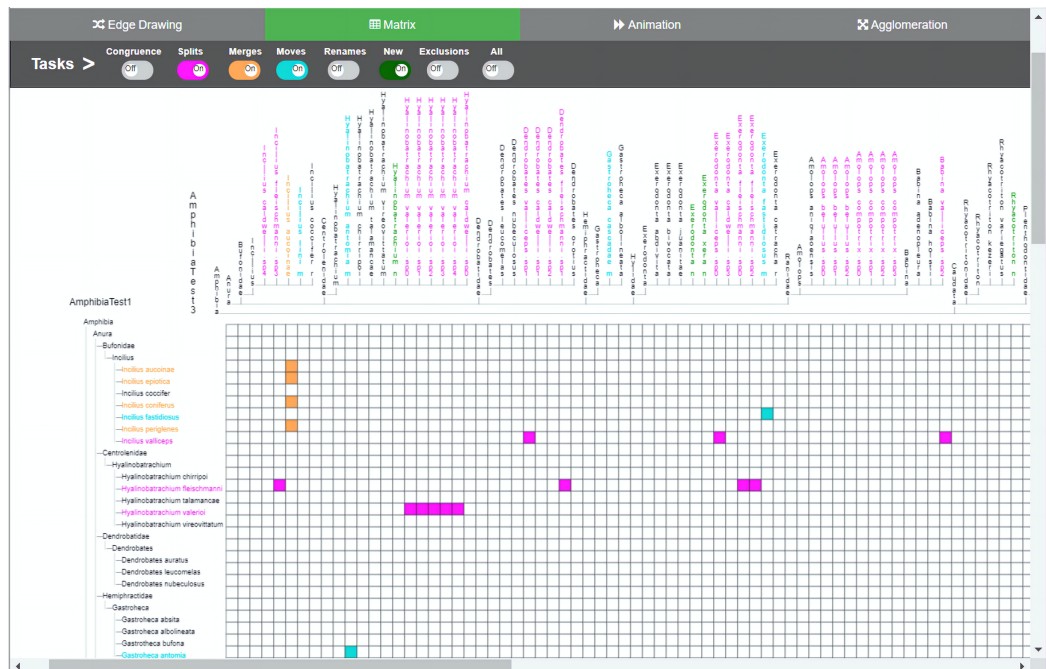

**Figure 4 Matrix method implementation.** The two taxonomies are arranged along the horizontal and vertical axes of the matrix. Changes between the two comparable taxonomies are highlighted thorough the color-coded cells. Through the colored toggle buttons on the menu, users can visualize either one or several types of changes at once. Users can select a specific taxon in order to visualize its changes.

software because many of the participants were located in different parts of the world. Figure 1 illustrates the user interface for the agglomeration method, Fig. 2 presents the animation method, Fig. 3 presents the edge drawing method and Fig. 4 presents the matrix method implementation. The two taxonomies to be compared, $T_1$ and $T_2$, are displayed as indented lists. Each method implementation is accessible by easily clicking on a tab. Users can inspect the data through the provided zooming features and by vertically scrolling for all methods. Additionally, horizontal scrolling is provided for *matrix*. The visualization layout is of course method-dependent. For *edge drawing* and *animation*, taxonomies are placed juxtaposed. $T_1$ is placed on the left side of the screen and $T_2$ on the right side. For *matrix*, taxonomy $T_1$ is also placed on the left side but $T_2$ is at the top of the matrix. Finally, for *agglomeration*, $T_1$ and $T_2$ are interleaved and centered horizontally.

The main menu is common to all methods, which adds uniformity to the user interface; however, the implementations of the different visualizations were optimized to best reflect the intrinsic characteristics of each method. The main menu is located at the top of the window and contains eight toggle buttons that display the changes induced by each type of curation task that we are considering, namely, *congruence*, *splits*, *merges*, *moves*, *renames*, *new*, *exclusions* and an additional *all* button. For example, when the *splits* switch is on, the visualization shows how each taxon with a split in $T_1$ is divided into taxa contained in $T_2$. The system is flexible enough to allow users to turn several buttons on at

the same time, in case they want to have several types of changes displayed simultaneously. For *animation* (see Fig. 2) additional controls to play and stop animations were added.

The color-coding scheme of the toggle buttons (i.e., tasks) is also the same across all methods and defines the types of changes to be visualized. We tried to use hues that were easily distinguishable to the eye in order to avoid confusion: blue for congruent taxa, pink for splits, orange for merges, light green for moves, light brownish purple for renames, red for exclusions, and green for added taxa. Familiar codes were also used; that is, red for exclusions and green for new taxa added. The representation of relations depends on the comparison method. For agglomeration, relations have to be inferred since data is interleaved and no explicit additional marks or lines can be included easily. Hence, for this method we decided to use an augmented color code in order to have a cue that would make it easier for participants to recognize to which taxonomy a node belongs and to highlight the types of changes between $T_1$ and $T_2$ (i.e., the relations between nodes). For this, we use the same hues but with different intensity, so the light nuanced nodes in the agglomerated structure indicate that they belong to the taxonomy of origin $T_1$ while the darker nuanced nodes indicate that they belong to the taxonomy of destination $T_2$. For example, a split of a taxon $x$ in $T_1$ into taxa $p$, $q$ and $r$ in $T_2$ would show $x$ in light pink color and $p$, $q$ and $r$ in a darker hue of pink. A legend was added to explain this color-coding. In the *agglomeration* method relations are permanent but not explicit.

In the *edge drawing* and *matrix* methods, relations are explicit and permanent. For instance, with the *edge drawing* method, a split of a taxon $x$ in $T_1$ into taxa $p$, $q$ and $r$ in $T_2$ is shown as three pink edges going horizontally from taxon $x$ to $p$, $q$ and $r$ in taxonomy $T_2$. In the *matrix* method, the same split case is shown as marked colored cells $(x; p)$, $(x; q)$ and $(x; r)$ respectively.

For *animation*, we considered two design choices: "animation by movement" and "animation by emergence". In the former, an animation consists of moving the target taxon from $T_1$ to its new position in $T_2$. In the latter, the target taxon would fade out from $T_1$ and would gradually appear in $T_2$. In either case, relations are explicit although temporary because they disappear when the animation is finished. We chose the first option because the paths followed by each moving taxon provides better traceability cues than the second one. Considering the split case described above, the animation would show $x$ moving towards taxonomy $T_2$. On its way, $x$ splits and disappears to let $p$, $q$ and $r$ appear and keep moving until each of them reaches its definitive position in $T_2$. The *animation* method per se does not necessarily involve leaving an explicit trace (as *edge drawing* does).

Two curation tasks do not involve relations between nodes in the alternative taxonomies, namely, *identify new taxa added* and *identify excluded taxa*. Given that inclusions and exclusions take place only in one of the taxonomies, the system visualizes these situations only in the taxonomy in which they occurred. Thus, excluded taxa are visualized in red color in taxonomy $T_1$ and included taxa in green color in taxonomy $T_2$. Without the use of color, asking users to visually infer which taxa were excluded from

**Table 2 Participants' profile.**

| Id | Area | Degree | Professional experience (years) |
|---|---|---|---|
| E1 | Botany, Bioinformatics | PhD | 32 |
| E2 | Botany, Forestry | MSc | 10 |
| E3 | Informatics, Bioinformatics | Engineer | 21 |
| E4 | Botany, Forestry | PhD | 28 |
| E5 | Botany | MSc | 15 |
| E6 | Botany, Bioinformatics | PhD | 31 |
| E7 | Botany, Forestry | Master | 21 |
| E8 | Botany | MSc | 21 |
| E9 | Botany | PhD | 23 |
| E10 | Biology | MSc | 30 |
| E11 | Ecology | PhD | 12 |
| E12 | Ichthyology | PhD | 32 |

$T_1$ and which ones were included into $T_2$ would require too much mental effort, especially when taxonomies are large.

The software included interactions that we considered crucial, such as navigating, selecting and zooming. For zooming, we took advantage of the browser's functionality and it is offered via three different alternatives: (a) by using the zoom feature on the browser's menu, (b) using the browser controls, that is, by pressing the <ctrl> key and moving the mouse scroll wheel up or down, or (c) pressing the key *I* on the keyboard for *zoom in* and the key *O* for *zoom out*. Functions such as undo/redo, edit and statistics were left out of the prototype since they were not crucial for the study. We developed the software incrementally through several iterations until we reached balanced implementations of the four methods. At the end of each iteration, computer science students tested the software. Tests were also conducted involving an experienced taxonomist and a PhD student in computer science.

## Participants

Twelve experts were recruited from our professional network. They received no compensation. Table 2 summarizes the participants' profiles. Each participant was given an identification number, ranging from E1 to E12. Eight of them are botanists (three of which are also Forestry engineers), two biologists (one entomologist and one ichthyologist), one ecologist, and one computing engineer (with 21 years of experience in biodiversity informatics). In addition, three of them reported Biodiversity Informatics as a second area of expertise. One participant holds an Engineering degree, five have a Master's degree and six have a PhD degree. Their average professional experience was 28 years and their average experience in the taxonomy field was 23 years; this includes taxonomic classification, taxonomy nomenclature, and curation of biological taxonomies. A total of 10 participants are male and two female. Three participants worked as full time university professors and the rest worked full time at herbaria, museums, or

biodiversity conservation initiatives. Participants came from three different countries and their expertise was with different taxonomic groups of organisms.

## Datasets

We carefully selected and designed the datasets, taking into account the level of familiarity participants might have with the data. Although taxonomists, in general, have extensive knowledge on certain groups of species, in practice, a taxonomist is only expert on a limited group of organisms; therefore, not all taxonomists need an overview of large subtrees. In addition, because of the large number and complexity of groups of species, their expertise is also geographically focused. Thus, despite of having ten botanists in our group of experts, all of them specialize in different groups of plants. In order not to favor any participant and avoid the eventual bias, we did not choose groups of species that were known by any of the experts. Therefore, we chose an unfamiliar taxonomy. It should not be very large since we did not want to burden participants by spending too much time performing the user study. However, at the same time, the dataset should be large enough to contain representative cases of all types of changes. We therefore used a small-size real taxonomy and derived artificial variants from it.

We downloaded a set of 66 species of amphibians from the Catalogue of Life website (http://www.catalogueoflife.org/) with a total of 96 nodes. We called this the *seed taxonomy* from which we derived variations on the datasets (derived taxonomies) to be used for each method. It was important that the data sets were different, but at the same time similar enough to be comparable. Therefore, we programmed an *artificial taxonomy generator* to which we input as parameters the percentage of splits, merges, movements, renames, additions and exclusions that we wanted to add to the seed taxonomy $T_1$. The generator randomly selected the taxa to introduce the changes and verified that only one change was introduced to each taxa to be modified. In this way, we prevented data conflicts, since more than one change to a taxon could generate inconsistent data. We also verified that, although questions were identical, the datasets would produce different answers for each method. The number of nodes in the derived datasets varied between 78 and 116 nodes. Table 3 describes the main characteristics of the four derived datasets, that is, the number of nodes, of species, of splits cases, merges, moves, renames, new and excluded taxa. The goal of this setup was to ask experts to visualize changes in four pairs of datasets: $(T_1, T_2)$, $(T_1, T_3)$, $(T_1, T_4)$ and $(T_1, T_5)$, with respect to *edge drawing*, *matrix*, *animation* and *agglomeration*, respectively. The derived datasets $T_2$, $T_3$, $T_4$ and $T_5$ are similar because they are all obtained from the seed taxonomy and have roughly the same number of changes. We avoided the use of the same pair of datasets across all visualization methods in order to neutralize a potential bias introduced by a learning effect.

## User study protocol

We planned the user study for a 2-hour session with each participant. During the session, participants would work with the interactive software environment to perform some exercises and to answer questions from a questionnaire. Seven out of the 12 experts lived overseas; therefore, the session was conducted remotely via a video call for them. For the

**Table 3 Derived datasets.**

|  | $T_2$ | $T_3$ | $T_4$ | $T_5$ |
|---|---|---|---|---|
| Nodes | 116 | 78 | 105 | 114 |
| Species | 86 | 55 | 75 | 84 |
| Splits | 6 | 6 | 7 | 9 |
| Merges | 7 | 3 | 5 | 3 |
| Moves | 6 | 4 | 4 | 6 |
| Renames | 7 | 4 | 6 | 6 |
| New | 6 | 4 | 3 | 6 |
| Excluded | 4 | 3 | 4 | 8 |

rest of participants, sessions were face–face. For participants in remote sessions, at the beginning of the session, we shared a link were the software and data were hosted. In case of the face–face interviews, we supplied a laptop computer. In both settings, access to the software environment was via web browser. We followed the same interview protocol for all participants.

A written guide and a 15 minute descriptive video of the software environment were available to the participants at least two days before the session, so that they could get familiar with it. Access to the software environment, datasets and questionnaire was not provided before the interview session.

A moderator was leading the session and assisted participants, while an observer was taking notes. Participants did not have to write down the answers; both the moderator and the observer would write the participants' answers on an answer sheet that they had previously prepared. Audios of the interviews were recorded for later confirmation of answers and analysis. At the beginning of the session, we checked to see if the participants had studied the guide and video beforehand and if they had any questions. In case they had not done so or if they needed to clarify any aspect, the moderator offered a demonstration of the software and resolved the doubts. Exercises were not started until both the participant and moderator felt they were ready; only then did the moderator provide the link for participants to access the interactive environment. Working speed was not to be measured and participants were made aware of the fact that they had no time limit to answer the questions and were able to express any inquiry, doubt or suggestion at any time. Participants were also asked and reminded to think aloud while solving the questions. Our goal was to get insights on how they carry out the data exploration and the tasks.

We designed an instrument that consists of twelve task performing exercises, nine method assessment questions and one open-ended comments section. The task performing exercises have clearly correct answers and were intended to measure the participants' effectiveness. The method assessment questions were intended to obtain participants' perception. The purpose of the open-ended question was to obtain additional feedback on user satisfaction and suggestions for a future design of an interactive visualization system. The study started with an exercise where participants had to identify

the most common type of change (overview task). Next, exercises were targeted to identify splits, merges, renames, moves, added or excluded taxa and ended with an overview question again. Each task-performing exercise had to be answered with each method. For instance, instructions such as "Use the Matrix method: Explore the visualization and find into what taxa *Babina caldwelli* was split?" were followed by the same question for all methods. However, the taxon to be used in each exercise (*Babina caldwelli*) was different for each method.

We randomized the order in which participants used each method on each question. Participants performed the exercises related to one task (for instance, identification of splits) and then were asked to assess each method to perform such task. The nine method-assessment questions consisted of five-level Likert scale items that assessed how good each method was to carry out the task. In the course of the session, participants had access to a copy of the questions and instructions, especially because taxa names were in Latin, and we wanted to avoid any confusion.

## Analysis

For the analysis of the results, we organized the participants' responses into a spreadsheet. We gathered three types of data: (a) the effectiveness data, that is, whether the participants answered each question correctly or incorrectly, (b) the user satisfaction data, that is, the Likert-scale ratings that participants gave to each method after accomplishing each task and (c) the qualitative data, that is, the thinking-aloud comments and the suggestions that participants provided during the session. Quantitative analysis was performed on data of types (a) and (b) by using a statistical package. We used non-parametric statistics with alpha = 0.05 and compared medians to determine that differences are not due to chance. For the analysis of effectiveness, we used the Cochran's $Q$ test, which can be used when you have a group of people performing a series of tasks where the outcome is dichotomic (e.g., success or failure). For the analysis of participants' satisfaction, we used the Friedman test, which is appropriate for within-subjects designs that have three or more conditions, and particularly it can be used for the analysis of ordinal data, such as the Likert-scale responses (*MacKenzie, 2013*). When necessary, both tests were followed by pairwise comparison using Dunn's test with Bonferroni correction.

For the qualitative analysis, we applied the following procedure. The responses were first placed in the same order in which the questions were presented to the participants, and, then, they were sorted by method. Two columns were designated for each participant, one to record their comments and suggestions (e.g., E1) and another one to afterwards register the codes generated during the qualitative analysis (e.g., E1-codes). Secondly, we listened to the audio recordings checking for additional feedback from the participants, which we added to the spreadsheet. Thirdly, we conducted a qualitative analysis: the first author made several coding passes using *open coding* (*Charmaz, 2006*) to obtain a first coding version that was then shared with the other authors. We coded participants' interactions and feedback. Repeated or related topics were grouped together, revised and re-grouped through several refinement cycles until we reached an agreement with twelve categories to finally conclude with four meaningful themes. During the process, we also

**Table 4 Comparison of participants' effectiveness using Cochran's Q test.** Each row corresponds to a task-performing exercise. Each cell of the row contains the results of the participants' effectiveness with each method for the evaluated exercise, as well as the chi-square and $p$-value resulting from the Cochran's Q test. Resulting less-effective methods for each exercise are shaded in orange.

| Exercise | Frequency (%) | | | | $\chi^2$ | $p$-Value |
|---|---|---|---|---|---|---|
| | Agg | Ani | Edg | Mat | | |
| (1) Which is the most common type of change? | 9 (75) | 11 (92) | 1 (8) | 11 (92) | 25.500 | 0.000 |
| (2) Into what taxa was taxon "t" split? | 11 (92) | 10 (83) | 11 (92) | 11 (92) | 3.000 | 0.392 |
| (3) Which species was split most? | 4 (33) | 7 (58) | 12 (100) | 11 (92) | 17.571 | 0.001 |
| (4) Was species "s" merged with any other species? | 11 (92) | 11 (92) | 12 (100) | 12 (100) | 3.000 | 0.392 |
| (5) With which other species was taxon "t" merged? | 12 (100) | 11 (92) | 12 (100) | 12 (100) | 3.000 | 0.392 |
| (6) Which is the new name of taxon "t"? | 12 (100) | 12 (100) | 12 (100) | 12 (100) | – | – |
| (7) Which was the previous name of "t"? | 11 (92) | 12 (100) | 12 (100) | 12 (100) | 3.000 | 0.392 |
| (8) Which species were moved to genus "g"? | 5 (42) | 12 (100) | 12 (100) | 12 (100) | 21.000 | 0.000 |
| (9) Which family has the most species added? | 11 (92) | 11 (92) | 12 (100) | 12 (100) | 2.000 | 0.572 |
| (10) Genus to which more than one species were excluded? | 11 (92) | 12 (100) | 12 (100) | 12 (100) | 3.000 | 0.392 |
| (11) What types of changes occurred to taxon "t"? | 6 (50) | 9 (75) | 11 (92) | 12 (100) | 9.692 | 0.021 |
| (12) Identify which genus shows most changes | 2 (17) | 7 (58) | 7 (58) | 7 (58) | 10.714 | 0.013 |
| Effectiveness (overall) | 105 (73) | 125 (87) | 126 (88) | 136 (94) | 40.480 | 0.000 |

organized the positive and negative comments, as well as the participants' suggestions for improving the methods.

# RESULTS

The study took 2:15 hours on average per participant. We first present quantitative results on participants' effectiveness and satisfaction, and then findings from the qualitative analysis.

## Effectiveness

The results of the participants' effectiveness on the task-performing exercises are summarized in Table 4. Overall results indicate that participants obtained more correct answers with *matrix* (94%), then with *edge drawing* (88%), followed by *animation* (87%) and then with *agglomeration* (73%). We tested for statistical significance by using Cochran's Q test for $N = 12$ and DF = 3. We did not find significant differences on participants' responses between pairs of methods (*matrix, edge*), (*matrix, animation*) and (*edge, animation*). However, we did find differences ($\chi^2 = 40.480$, $p$-value = 0.05) between agglomeration (73%) and the other methods, meaning that participants were less effective with the agglomeration method.

We also did a quantitative analysis on responses to each exercise. We did not find significant differences among participants' responses when identifying: (a) into which taxa a taxon was split (exercise 2), (b) whether species were merged and how (exercises 4 and 5), (c) whether species were renamed (exercises 6 and 7), (d) whether any species were added to a version of the taxonomy (exercise 9) and (e) whether any species were excluded (exercise 10). We found significant differences in participants' responses in identifying:

(a) an overview of changes (exercises 1 and 12), (b) which species were most divided (exercise 3), (c) moved taxa (exercise 8) and (d) all changes on a taxon (exercise 11). For these cases, a post hoc pairwise comparison was performed in order to determine where the differences occurred. Resulting less-effective methods are highlighted in orange color in Table 4:

- Exercise 1. Overview of changes. We found differences ($\chi^2 = 25.500$, $p$-value $< 0.05$) between the following pairs of methods: (*animation*, *edge drawing*), (*matrix*, *edge drawing*) and (*agglomeration*, *edge drawing*). This indicates that the effectiveness with *edge drawing* (8%) was lower with respect to *agglomeration* (75%), *animation* (92%) and *matrix* (92%).

- Exercise 3. Identification of splits. We found differences ($\chi^2 = 17.571$, $p$-value $< 0.05$) between pairs of methods (*edge drawing*, *agglomeration*) and (*matrix*, *agglomeration*). These results indicate that the effectiveness with *agglomeration* was different with respect to the other methods. Participants were less effective with *agglomeration* (33%) and more effective with *edge drawing* (100%) and *matrix* (92%).

- Exercise 8. Identification of moved taxa. We found differences ($\chi^2 = 21.000$, $p$-value $< 0.05$) between pairs of methods (*animation*, *agglomeration*), (*edge drawing*, *agglomeration*) and (*matrix*, *agglomeration*). This indicates that participants were less effective with *agglomeration* (42%) whereas they were more effective with the other methods (100%).

- Exercise 11. Focus on a taxon. We found differences ($\chi^2 = 9.692$, $p$-value $< 0.05$) between the pair of methods (*agglomeration*, *matrix*). This indicates that participants were less effective with *agglomeration* (50%) and more effective with *matrix* (100%).

- Exercise 12. Overview of changes. We found differences ($\chi^2 = 10.714$, $p$-value $< 0.05$) between pairs of methods (*animation*, *agglomeration*), (*edge drawing*, *agglomeration*) and (*matrix*, *agglomeration*). This indicates that participants were less effective with *agglomeration* (17%) than with *animation* (58%), *edge drawing* (58%) and *matrix* (58%).

### Satisfaction level

Right after carrying out the task-performing exercises for each task, participants answered a Likert-scale questionnaire to assess the methods. The questions had the following structure: "How good do you think each method is in order to perform task *t*? For each method provide a rating between 1 and 5, where 1 stands for "poor", 2 for "fair", 3 for "good", 4 for "very good" and 5 for "excellent"". We performed a statistical analysis on the participants' ratings using the Friedman test. Table 5 summarizes the results for $N = 12$ and DF = 3. We did not find any difference in participants' responses to accomplish the task for the identification of excluded species (question 10e). Neither we found differences regarding the identification of added species (question 9e) after running the post pairwise comparison. On the contrary, we found differences in participants' responses to carry out tasks for the identification of the most common type of change (1e), splits (3e), merges (5e), renaming (7e), moves (8e), changes to a taxon (12e) and the general

**Table 5 Comparison of participants' satisfaction using Friedman test.** Each row corresponds to a task-performing exercise. Each cell of the row contains the results of the participants' satisfaction level with each method for the evaluated exercise, as well as the chi-square and $p$-value resulting from the Friedman test. Higher satisfaction level is shaded in green and less satisfaction level in orange.

| Question How good do you think is each method in order to … | Median | | | | $\chi^2$ | $p$-Value |
|---|---|---|---|---|---|---|
| | Agg | Ani | Edg | Mat | | |
| 1e- … identify the most common type of change? | 2.08 | 1.79 | 3.29 | 2.83 | 12.588 | 0.006 |
| 3e- … identify splits? | 1.50 | 1.50 | 3.79 | 3.21 | 33.055 | 0.000 |
| 5e- … identify merges? | 2.00 | 1.71 | 3.71 | 2.58 | 20.050 | 0.000 |
| 7e- … identify renaming of taxa? | 2.13 | 1.50 | 3.67 | 2.71 | 21.559 | 0.000 |
| 8e- … identify moves? | 1.54 | 1.88 | 3.83 | 2.75 | 24.295 | 0.000 |
| 9e- … identify new species added? | 2.75 | 2.63 | 2.96 | 1.67 | 9.539 | 0.023 |
| 10e- … identify excluded taxa? | 2.67 | 1.88 | 2.79 | 2.67 | 7.062 | 0.070 |
| 12-e- … identify changes to a taxon? | 1.88 | 1.79 | 3.46 | 2.88 | 16.057 | 0.001 |
| 13- … visualize differences and similarities between two taxonomies? (Overall) | 2.97 | 2.84 | 4.97 | 4.09 | 25.064 | 0.000 |

methods assessment question (13). The post hoc pairwise comparison gave the following results (in Table 5, higher satisfaction level is shown in green and less satisfaction level is highlighted in orange):

- Question 1e-overview ($\chi^2$ = 12.588, $p$-value = 0.05). We found differences between pairs of methods (*animation, edge drawing*). Participants gave a better rating to the *edge drawing* method (median = 3.29 ~ "good/very good") than *animation* (median = 1.79 ~ "fair").

- Question 3e-splits ($\chi^2$ = 33.055, $p$-value = 0.05). We found differences between pairs of methods (*agglomeration, matrix*) (*agglomeration, edge drawing*) (*animation, matrix*) and (*animation, edge drawing*). There was no difference between *agglomeration* and *animation* and neither *matrix* and *edge drawing*. Participants ratings for *agglomeration* (median = 1.50 ~ "poor/fair") and *animation* (median = 1.50 ~ "poor/fair") were the lowest while for *matrix* (median = 3.21 ~ "good") and *edge drawing* (median = 3.79 ~ "good/very good") were the highest ones.

- Question 5e-merges ($\chi^2$ = 20.050, $p$-value = 0.05). We found differences between pairs of methods (*animation, edge drawing*) and (*agglomeration, edge drawing*). Participants assessed the *edge drawing* method with the highest rating (median = 3.71 ~ "good/very good") compared to *agglomeration* (median = 2.0 ~ "fair") and *animation* (median = 1.71 ~ "poor/fair").

- Question 7e-renames ($\chi^2$ = 21.559, $p$-value < 0.05). We found differences between pairs of methods (*animation, edge drawing*) and (*agglomeration, edge drawing*). Participants assessed the *edge drawing* method with the highest rating (median = 3.67 ~ "good/very good") than *agglomeration* (median = 2.13 ~ "fair") and *animation* (median = 1.50 ~ "poor/fair").

- Question 8e-moves ($\chi^2$ = 24.295, $p$-value = 0.05). We found differences between pairs of methods (*agglomeration, edge drawing*) and (*animation, edge drawing*). Participants

assessed the *edge drawing* method with the highest rating (median = 3.83 ~ "very good") than *agglomeration* (median = 1.54 ~ "poor/fair") and *animation* (median = 1.88 ~ "poor/fair").

- Question 12e-focus ($\chi^2$ = 16.057, *p*-value = 0.05). We found differences between pairs of methods (*animation*, *edge drawing*) and (*agglomeration*, *edge drawing*). Participants assessed the *edge drawing* method with the highest rating (median = 3.46 ~ "good/very good") than *agglomeration* (median = 1.88 ~ "fair") and *animation* (median = 1.79 ~ "fair").

- Question 13-general assessment of all methods. ($\chi^2$ = 16.057, *p*-value = 0.05). We found differences between pairs of methods (*animation*, *edge drawing*) and (*agglomeration*, *edge drawing*). Participants assessed the *edge drawing* method with the highest rating (median = 4.97 ~ "excellent") than *agglomeration* (median = 2.97 ~ "good") and *animation* (median = 2.84 ~ "good").

It is important to notice that response for question 13 summarizes the participants' level of satisfaction.

### Findings from qualitative analysis

Regarding *edge drawing*, all participants referred to this method throughout all exercises with expressions such as: "easy", "very direct", "I can easily relate taxa", "it is very fast", "I do not have to think too much" and "you can see … at a glance". One participant (E6) said that it was the best because "you can clearly see the origin and the destination". Another participant (E7) considered that *edge drawing* "is familiar, it is similar to an "associate" type of exercise".

Participants' feedback on *matrix* highlighted this method as good for the visualization of general overviews and the identification of patterns, eight participants mentioned it (E2, E3, E7, E8, E9, E10, E11 and E12). Five participants (E6, E7, E10, E11 and E12) mentioned that *matrix* was the fastest one. Another participant (E2) recognized that "it is easy to see in a row the changes to a taxon". Three participants (E10, E11 and E12) considered that the required vertical and horizontal scrolling add complexity and two other participants (E2 and E10) mentioned that scaling could be a problem. One participant (E1) complained that he had to use his fingers on the screen to follow the relations in the two dimensions. Eleven participants complained about the vertical name implementation in the top hierarchy (all except E2), and two participants (E1 and E10) found navigation difficult because parts of the hierarchies were off the screen.

Regarding *animation*, two participants (E1 and E8) rated it positively indicating that it was "dynamic", and therefore "fun". However, eight out of the 12 participants described this method in negative terms such as "difficult", "ineffective", "hard to follow", "complicated", "not intuitive" and "waiting until the end of the animation is a waste of time". Five participants considered that the *animation* was not necessary. Participants emphasized that changes between the two taxonomies were very difficult to follow because they could very soon forget what happened, especially if taxonomies were large. Five participants indicated that, while taxa were moving, it was easy to lose track of the relation

between origin and destination because the taxa were moving. Most participants speeded up the animation, giving the impression that they wanted it to get to an end quickly, but some had to execute it several times before being able to solve the exercise.

Participants indicated that it was very difficult to carry out the tasks with the *agglomeration* method, except for the identification of excluded taxa. Eight participants (E2, E5, E7, E8, E9, E10, E11, E12) referred to *agglomeration* as very good when looking for specific taxa or to focusing on a small part of the taxonomy. However, all participants also described it in negative terms, such as "difficult", "very complicated", "requires too much effort", "not evident", "confusing" and "very difficult to know origin and destination". Participants complained that this method involved many variables that were difficult to remember (that is, many color hues) and that it required considerable effort to recognize differences. However, two participants thought that the *agglomeration* view could be complementary to *edge drawing*, and that it could work well for small taxonomic groups.

We coded and organized the participants' feedback until we reached themes that we considered meaningful. Our observations show four specific issues that are relevant when performing tasks for the curation of biological taxonomies:

- **Explicit representation of changes.** Changes are visualized through relations among taxa. Nine out of twelve participants clearly indicated that being able to identify the origin and destination of relations was very important to recognize changes when comparing biological taxonomies. Participants' suggestions such as; "add edges to animation", "add edges to matrix", or "add numbers to each change in the agglomeration method" are indications that they would prefer to see relations explicitly and, therefore, prefer methods that explicitly represent the changes.

- **Efficiency.** Participants often commented about speed and time needed to solve the exercises. Across exercises, they referred to the importance of understanding what is going on at a glance. They expressed feeling frustrated when having to wait for the animation to end. They speeded up the animation when they felt that solving the exercise was taking too much time. Participants considered that having to scroll horizontally and relate rows and columns of the matrix or having to interpret different colors as in agglomeration were steps that consumed time.

- **Multiple views.** Several participants commented that the methods could be complementary; for instance, that the *edge drawing* and the *matrix* methods could be used to visualize all cases at once whereas the *animation* and the *agglomeration* methods could be useful when analyzing smaller groups of species. They explained that, by combining several methods, the advantages of one method could overcome the disadvantages of another one. On the other hand, the experts also emphasized the convenience of having both overview and detailed views; the first one to obtain a general understanding of changes and the second one to obtain detailed information on a focused part of the taxonomies.

- **Visual and numerical summaries.** When asked for number of taxa that match a certain condition, participants expressed their frustration because they had to count manually

and suggested to add statistics to the software environment. Although obtaining statistical information is one of the tasks for the curation of biological taxonomies, we decided to leave it out of this study on purpose in order to force participants focus on the visualizations. The intention of quantity-related questions was to see if participants were able to visually identify magnitude of changes (for instance, matrix resulted good). We obtained confirmation on the importance of providing numerical understanding of changes.

## Suggested improvements

The methods that received most suggestions for improvement were the ones that had the lowest participants' effectiveness and preference. Suggestions for agglomeration focused on mechanisms that would make the relations explicit somehow and allow them to be recognized quickly. For instance: (a) add numbers instead of different color hues to indicate taxa of the origin and taxa of the destination, (b) use a different color hue for each change (not only for each type of change), (c) instead of using several color hues, consider glyphs or some other visual cues, (d) color the background of the text instead of the text, (e) visually separate the taxonomies on user's demand, (f) separate the legend so that hues associated to origin are placed on the left side and the hues associated to the destination are placed on the right side of the screen. For animation participants provided suggestions such as: (a) add a time slider and a rewind button, (b) identify each change with a number, (c) add traces as in edge drawing, (d) identify each specific change with a different color, (e) maybe consider the use of animation for comparing a small part of the taxonomies. For matrix, participants also made suggestions to improve the visualization of relations; for instance, (a) add a feature to freeze rows or columns in place and ease the visualization of relations when vertical or horizontal scrolling is needed, (b) add horizontal and vertical guiding lines to ease following the relations, (c) add a colored rectangle around the excluded or added taxa in order to highlight these changes, (d) add edges in order to make the relations more explicit, (e) consider the matrix method as a way to feed a database with the relations between the two taxonomies. Suggestions for edge drawing included: (a) use more intense color hues and (b) provide features that ease the comparison between taxa of higher taxonomic ranks (for instance, at the family level). Three participants (E3, E5, E6) mentioned on several occasions that the identification of relations was easier when the involved taxa were closer together, within the same view. They expressed this thought as they were solving matrix and agglomeration exercises. On the other hand, four participants (E3, E6, E10, E12) mentioned that vertical scrolling was fine in edge drawing, since "it is very familiar". Feedback obtained from the open-ended comments section of the questionnaire also included suggestions for enhancements to the implementation of the methods. Regarding the representation of hierarchies, participants' suggestions included the elimination of lines that indicate hierarchical structure and to use only indentation as a visual cue to recognize hierarchy (some participants believed that the visualization could look cleaner). They also mentioned that the visual clutter caused by long names might be overcome by using abbreviations when possible (for instance, for the genus part of the species names).

## DISCUSSION

The research question we investigated was how well each of the four methods of hierarchy comparison supports the tasks of contrasting two versions of a taxonomy. The quantitative and qualitative results revealed differences among the methods. The difference in effectiveness occurred only with respect to agglomeration, as the participants were the least effective with this method. One likely explanation is that in all methods, the changes between the two taxonomies were consistently represented throughout predetermined colors for each type of change, but in agglomeration each change was represented by two tonalities of the predetermined color, one to indicate the taxon of origin and the other to indicate the taxon of destination. This might have added complexity that affected the participants' effectiveness with this method. Overall, participants were very effective with the other three methods, which might be because the participants could take as much time as they considered necessary to solve the exercises. The number of correct answers with animation and with edge drawing was quite similar (125 and 126 respectively). In spite of this similarity in the participants' accuracy, and that both methods used juxtaposed layouts, the user satisfaction results indicate greater participants' preference for edge drawing. Comparing the amount of correct answers, participants showed similar performance in many exercises with edge drawing and matrix. The difference between these two methods comes mainly from the responses to the overview exercise where they had to identify "(1) Which is the most common type of change?" where only one participant answered correctly with edge drawing and 11 participants with matrix. This might indicate that matrix works well to get a general overview of changes. This is reflected also in the participants' feedback when they highlighted that this method was good for pattern recognition.

The effectiveness on recognizing new taxa or excluded taxa was similar with all methods. This is explained by the fact that both new and excluded species are visualized only in one of the taxonomies, and require no relations between the involved taxonomies.

In another respect, we noticed that participants were more effective at identifying changes at the lowest level of the taxonomy (i.e., species level) than when trying to recognize changes at upper levels (such as at the genus level, exercise 12). This might suggest the convenience of having summary overviews on changes at higher-level taxa.

Both the quantitative and the qualitative analysis coincide that the agglomeration method ranked last. In spite that the results on effectiveness did not show clear differences among animation, edge drawing and matrix, both the participants' feedback on satisfaction and the qualitative findings suggest edge drawing in the first place.

### Threats to validity

Various factors can limit the validity of the results of visualization user studies, including the number of participants, the choice of the datasets, the design of the study, and the data analysis.

The study included twelve expert users. A small number of participants can affect the quality of results, especially with respect to quantitative results. However, with respect to qualitative findings, the number was sufficient. After completing a first set of nine

interviews, we recorded the data on a spreadsheet and did a preliminary processing. Afterwards, when we finished all 12 interviews, we noticed that the qualitative results repeated (that is, with 33% more participants than in the preliminary processing). Such consistency is an indication of dependability regarding the qualitative results.

Despite the task randomization and the careful selection of data, a learning bias might have been introduced. Learning effects are a risk to all small-scale studies as randomization only partly counter balances for small numbers and individual opinions of participants might have been influenced by the sequence of tasks.

A restriction of the study is the small size of the datasets, which contained between 78 and 116 nodes. As the data were unknown to all participants, the datasets had to be restricted to a small size in order to carry out the study in reasonable time. Still, the datasets contained all types of required changes to perform the tasks. In the case of the dataset for matrix, it turned out a little smaller than the other ones, but large enough to allow users to experience both horizontal and vertical scrolling, so we considered it was fine for the study; however, the difference in size compared to the other datasets might have added some bias to the study. For visualizing larger excerpts or whole taxonomies, other visualizations might be necessary, and our results may not generalize to this. Also, the binding of datasets to methods could have introduced a slight confounding factor but seemed an acceptable trade-off, which we took to simplify the evaluation logistics.

Although the transferability of results is limited by (a) the specific domain application, (b) the tasks studied and (c) the data sets, the comparison of hierarchical structures is independent of the application domain; thus, some features of the study might contribute to other contexts where users need to identify divided, joined, moved or renamed nodes between two hierarchies.

The within-subject design is applied in studies with a small number of participants. All participants interacted with every method. In this way, we expected to reduce biases associated with individual differences. However, the within-subject design may bias participants because of the carry over effect; that is, once participants perform a task with one method, they may expect certain conditions to happen in the next method to evaluate. We tried to counterbalance bias from the learning and tiring effects by asking participants to interact with the methods in a randomly established order. In addition, although the questions were identical, the data sets would produce different answers for each method in order to prevent the learning effect.

A bias may be introduced by the design of the study or the design of the interactive software environment. We tried to implement the essence of the methods as well as to keep a standard user interface for all visualizations (same main menu and color codes throughout all methods), but limitations may come from the design choices and implementation decisions. For instance, zooming and scrolling features were limited and not designed for very large amounts of data; also, participants were unsatisfied regarding the vertical text orientation of the matrix implementation. Applying coloring was more difficult in the agglomeration approach because, in this method, color is also required to indicate the different versions of the taxonomy. We decided for a mixed brightness and

color encoding as a tradeoff, which seemed introducing a smaller bias, instead of avoiding the use of color for encoding types of changes or introducing an inconsistent use of color across the methods.

We aimed to objectively examine the collected data. For the quantitative part of the study, we used statistical tests to analyze if the differences between the medians were significant. For the qualitative part of the study we carefully organized the data and coded the participants' feedback and interactions through several refinement cycles. By counting and grouping similar feedback from participants, we were able to define the codes and themes. Nonetheless, the interpretation of the data may be subject to the perspective of the researchers.

## Implications

The participants performed well with edge drawing and consider it, in general, the best method; despite they did not have the best performance with it. For overview tasks, participants showed similar effectiveness with matrix and animation, however they preferred matrix.

Results indicate that identifying explicitly the origin and destination of taxa is very relevant for a more efficient identification of changes; edge drawing and matrix methods seem to have facilitated it. The participants' need to determine origin and destination may explain that edge drawing outperformed animation in both effectiveness and user satisfaction. Both methods present the taxonomies in a juxtaposed layout, however, relations are not explicit in animation. Some participants considered that animation could be useful to focus on changes in small taxonomic groups, which reaffirms the scope of animations, as indicated by *Graham & Kennedy (2007)*.

During the sessions, we noticed that sometimes taxonomists wanted to see the big picture and then focus on a smaller group of organisms of their interest. Also, sometimes they wanted to go directly to the group they want to inspect. Thus, future research should consider easily toggling between overview and detailed views as well as search and filtering functions. Text (that is, taxa names) is crucial when comparing taxonomies. Visual cues such as color, size, shapes or glyphs are not enough to recognize the differences and similarities. Text must be legible. The users would have to read names, which would need to be accommodated efficiently avoiding cluttering. Unlike other studies, we were not assessing how participants use the tool (*Lutz et al., 2014*), neither we were measuring the prototype efficiency for comparing hierarchies (*Graham & Kennedy, 2010*). Instead, our contribution lays in the assessment of the four visualization methods for the comparison of pairs of biological hierarchies with respect to curation tasks.

A final remark about methods and tools for the comparison of biological taxonomies is that the methods we evaluate in this work are those that have been mainly discussed in the scientific literature; however, their implementation in tools is limited. The Taxonomic Tree Tool (*Contian et al., 2016*) is, as far as we know, one of the few tools available online that allows us to explore relationships (such as ancestor-descendant relationships) when comparing two biological taxonomies. In a post-study contact with some participants,

we asked about the software they use for taxonomy comparison. We obtained five responses and only one indicated to occasionally use the Taxonomic Name Resolution Service (http://tnrs.iplantcollaborative.org) to compare a list of plant names with an authoritative database of published names. We do not know the precise reasons why the methods are not in widespread use, but some possibilities could be that the developments only reached prototype versions and are not available online, that the visualizations could be too complex, or the difficulty that comes with the lack of standardization of data in different communities.

## CONCLUSIONS AND FUTURE WORK

This study contributes insights on the capacity of four visualization methods for hierarchy comparison in typical biological taxonomy curation tasks. Twelve expert taxonomists took part in a study and provided feedback. We performed quantitative as well as qualitative analysis. The results clearly show differences among the methods, on both users' effectiveness and satisfaction: the edge drawing method was preferred over other methods.

In this study, the data sets were selected to avoid bias, all participants used the same datasets, and participants were able answer the questions in reasonable time. However, it will be interesting to design a similar study with larger datasets. Another approach would be to design a study in which the data would be specific and familiar to each participant.

Enhancements such as providing multiple views, adding visual cues at inner taxonomic rank levels, and avoiding overloading caused by long names and hierarchical structure lines, are insights for future research. Functions for searching, statistics and queries to retrieve the information of a taxon will be considered in a future design of a visualization environment. We also plan to further research on visual summary views to facilitate the comparison at different taxonomic rank levels. Finally, it will be important to make biological taxonomy comparison visualization tools available for use. This implies overcoming challenges such as promoting the standardization of data to facilitate data sharing and comparison.

## ACKNOWLEDGEMENTS

The authors would like to thank all expert taxonomists who collaborated with this research, dedicated the time to participate and provided very valuable feedback. We would also like to thank Eros Hernández-Romero and Ronald Andrés Bolaños-Rodríguez for their collaboration during the software development and the interview sessions, and Shivam Agarwal for his collaboration as test participant.

### Funding

This work was supported by the Costa Rica Institute of Technology (doctoral scholarship and research grant VIE-1370003). The funders had no role in study design, data collection and analysis, decision to publish, or preparation of the manuscript.

## Grant Disclosures

The following grant information was disclosed by the authors:
Costa Rica Institute of Technology: VIE-1370003.

## Competing Interests

The authors declare that they have no competing interests.

## Author Contributions

- Lilliana Sancho-Chavarria conceived and designed the experiments, performed the experiments, analyzed the data, performed the computation work, prepared figures and/or tables, authored or reviewed drafts of the paper, and approved the final draft.
- Fabian Beck conceived and designed the experiments, analyzed the data, authored or reviewed drafts of the paper, and approved the final draft.
- Erick Mata-Montero conceived and designed the experiments, analyzed the data, authored or reviewed drafts of the paper, and approved the final draft.

## Data Availability

The software, the data, the questionnaire, a written guide to the tool, a link to a demo video, and the analysis materials are available at GitHub: https://github.com/lsanchoc/MethodsTasksUserStudy.

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

**Von Carl L, Friedrich GJ. 1767.** *Systema naturae per regna tria naturae: secundum classes, ordines, genera, species, cum characteribus, differentiis, synonymis, locis*. Lipsiae: Impensis Georg. Emanuel. Beer.