# Peer review of "An expert study on hierarchy comparison methods applied to biological taxonomies curation"

_PeerJ Computer Science, doi:10.7717/peerj-cs.277_

## Round 0.1 · original submission · Minor Revisions

The reviewers have identified a number of concerns that should be addressable via revisions to the text. Please address all reviewer comments, paying attention to the clarification of some of the methods and assumptions (as raised by two reviewers, but primarily reviewer 1), the presentation and contextualization of results (reviewers 2 and 3), and other issues regarding the description of the system and the experiment (reviewer 3). The comments about experimental design, hypotheses, and possible confounding effects are particular concerns regarding the validity of the results. Please also consider Reviewer 2's comments regarding the graphical presentation of the results.

Reviewer 1 ·

Basic reporting

The paper as a whole is well structured and entirely readable (I spotted two typos, at Line 637 ‘curing’ should be ‘comparing’, Line 725 ‘legibly’ should be ‘legible’). The introduction gives adequate coverage to both the taxonomic problems/tasks and the visualisation techniques that are plausible solutions. The literature is well covered – occasionally a citation in the text is missing a year (e.g. for Lin et al’s web-based tool). There is only one figure and unfortunately it’s heavily pixelated – I can tell which of the four scenarios is which but the detail in all is obscured – this needs to be redone. Both the raw study results data and the test visualisations and example taxonomies used have been made available via a github account.

Experimental design

The research question is which visualisation is best suited to display the typical relationships between a pair of trees (taxonomies), where the mapping is not always a simple one-to-one relationship, but can involve renamings, splits and aggregations, exclusions and new additions. This is obviously relevant and meaningful to the taxonomists that must frequently understand and occasionally create such relationships between taxonomies.

The methods are described with sufficient detail, in conjunction with the test visualisations and taxonomies, to recreate the study, and the study itself is carried out with a reasonable number of participants (12) and with tasks that representative users consider important. The experiment design of recording user accuracy in performed tasks (along with audio recording), followed by Likert-style user preference questionnaires and open-ended questions is standard and therefore not a weak point. Details on how long each stage took and the sequencing of the test are supplied.
It appears from the text that rather than have users perform all the questions on each visualisation in turn, all the visualisations were tested on each question in turn? (The order of the visualisations was randomised for each question to avoid learning effects.) It’s unusual in that I haven’t seen it done like this before, but that isn’t the same as saying it’s worse. It would be good if the authors mentioned why they took this particular approach.

There are a couple of possible design flaws in the study. Firstly the study assigns a base taxonomy and one of four artificially generated taxonomies per visualisation condition. Even though the differences between the four generated taxonomies are small, the fact that one is always tied to a particular visualisation type means it could be a slight confounding factor i.e. how much difference in user performance was due to Visualisation A being better than B, or because Visualisation A always used taxonomy 1 and Visualisation B always used taxonomy 2? The authors do pick up on this in their results analysis, but think it maybe because the artificial dataset always associated with the matrix visualisation was slightly smaller, whereas the real problem is that this dataset is always associated with the matrix visualisation. It would have been perhaps preferable to cycle through combinations of visualisation type and artificial dataset when presenting the users with their tasks, rather than just vary the ordering of the visualisation type.
Secondly, with the agglomeration visualisation, colour is overloaded by being used for both the type of relationship a taxon is involved in between two taxonomies, and for indicating which taxonomy it belongs to. The authors again themselves acknowledge this later on.

One other choice I’m unsure about which isn’t revealed until the results stage are the labels for the Likert-type scale. The question is termed “How good do you think each method is in order to perform task t?”, whereas I think ‘effective’ would have been a better adjective than ‘good’. It leads to the unfamiliar situation where the middle term in the five-point scale, which I’d normally expect to be neutral (i.e. ‘average’ or ‘moderate’) is in fact labelled ‘good’.

Validity of the findings

The statistics used to investigate the data gathered from the user study appears to have undergone careful consideration – appropriate non-parametric tests for the Likert-style data and an appreciable effort made in coding the open-ended questions.

In the results section, the matrix condition isn’t mentioned in the last five of the satisfaction level bullet points (line 500-519) which seems curious – is this on purpose or a mistake?

The coding of the open questions / feedback reveals four categories and importantly details about why the users have these objective and subjective preferences e.g. animation and scrolling were seen as frustrating / inefficient so impacted on the impression of those visualisations that relied heavily on those.

The authors’ final findings in the discussion are that while effectiveness was difficult to distinguish between visualisation condition, the users had a clear subjective preference for the edge drawing method. This is something that often occurs in visualisation experiments, there is often little or no difference in performance between alternatives, but very clear preference.

The conclusions are clearly stated, with the clearest outcome being the subjective preference for the edge drawing condition, and ideas for future improvements to the study are made, along with recognition of some of the flaws of the current study.

Additional comments

Possibly, one of the more interesting points within the paper that the authors didn’t detail much was the generation of the artificial data sets. They describe using an artificial taxonomy generator (Line 333) that works on a seed taxonomy. It would have been interesting to see more information on this as artificial data generation is an important topic within data science at the moment and has been touched on in the visualisation field previously, especially for graphs/networks.

In short, there are issues with some of the initial experimental design choices (1.colour in agglomeration, 2.non-neutral middle value in questionnaire, 3.fixed pairing of data to visualisation type), that could be considered to threaten the validity of the results. However, the analysis itself is thorough and the rest of the paper to my mind is sound and doesn’t need much change apart from the points in the review above. I've marked this down as a major revision simply because I'd like to see the changes, even though the amount of text

So what the authors need to do is
1. convince that these design choices do not significantly alter the outcome of the study – in essence counter their own findings in the “threat to validity” section - i.e. would the agglomeration condition still have been difficult to use if marking the two taxonomies in a different way (glyphs, font style etc), given that the distinction between the two taxonomies is always going to be marked with a much less powerful perceptual variable than in the edge drawing, animation or matrix conditions (position getting replaced by line thickness (bold text) or glyph (shape))? Beyond this the alternative is to re-run the study which is obviously a much, much larger undertaking.
2. Generate a much clearer Fig 1.

Reviewer 2 ·

Basic reporting

Please see General Comments

Experimental design

Please see General Comments

Validity of the findings

Please see General Comments

Additional comments

This paper describes an experiment to understand differences among different visualization designs for tasks in understanding differences between a pair of biological taxonomies. The paper provides a rare example of a true expert level study for these kinds of tools. While the findings are not surprising, the mere fact that the authors have followed through on such a student should be commended, and provides a contribution to show that it is possible.

I feel this paper has a number of problems with exposition that, if fixed, would make it considerably stronger. On one hand, the paper is good enough that I was able to get the message and be convinced of its value. On the other hand, this took a little more effort, and I may not have gotten the full impact of the results.

1. (small) In the abstract and introduction: be clear the degree to which this is understanding existing designs. This is a little hard because while the broad categories of the designs have been presented, their application to the specific task and their specific implementation do matter.

2. (big) The paper is very cautious to talk about its narrow domain goal. This is refreshing - the authors are cautious not to over-claim. However, the paper never helps me understand the specific challenges of the domain. What is unique about this domain that it requires special consideration that may not apply more broadly? I actually think the insights of this paper may extend more broadly.

3. (small) The introduction made me wonder the degree to which there were good tools for the specific problem (or the general one). Other than research prototypes, have things made it into actual practice? This issue is addressed a little in the related work section, however, even there I am left feeling like the design of tools for the specific challenges of biological taxonomy comparison have not been addressed yet.

4. (small) Around line 56 - I feel like the writing is conflating documentation with the data. This could be made clearer.

5. (big) Around 71-72, the table, and other places in the paper: I feel the paper could better connect the tasks and categorization into existing task frameworks (e.g., Munzner, Gleicher, Schulz et al). Now, the categorization seems arbitrary and unmotivated. It also doesn't help that the categories are only explained later.

6. (small) I misread things such that I didn't realize the whole paragraph at 74 was talking about the pattern identification task (and that the later highlights were kinds of patterns to identify - except then the ideas of summary and overview are introduced). This paragraph is mixing ideas together.

7. (big) @107 - why aren't the methods in use? if they aren't in use, why do we believe they are worthy of study?

8. (big - general comment with small specific) @125 - One experimental design choice the authors made was to create their own implementation of all the methods. This has pros and cons (the methods are uniform, but they may not all be best-of-breed) that should be discussed. Also, the uniformity may not be fair (@128): it could be that some designs make it easier to add interactive features. I don't think this is a problem: but the experimental design choice should be more fully discussed.

9. (small, but connects to big #5) @169 - it would be nice to connect these frameworks to the present work, rather than just citing them.

10. (small) @215 - maybe this is a wording thing, but quantifying subjective response is still subjective

11. (big) The format of the manuscript placing the figures so far away exacerbates the problem that there are not enough of them: please give more an better figures to help the reader. In particular, the features of the program are hard to understand as described as text, and the numerical results are hard to keep track of without graphs.

12. (big/small, connects to #2) As I understand it, a problem in taxonomy comparison is that the taxonomies are huge and have many unfamiliar parts (even to experts). While this discussed as a limitation, it would be useful to know how useful/realistic the kinds of scales considered in the experiments are.

13. (huge) The experimental results only presented in as numbers text are impossible to appreciate. Please present them as a tables and figures. (standard practice is to provide both). It's ironic that a paper on visual comparison provides no visual comparisons to help the reader.

14. (small/big) The paper never really states the hypotheses for the experiments, which makes interpreting the results (what did they show?) a bit harder. Post-hoc, it is wrong to introduct hypotheses, but it might be good to phrase the research questions in the design and results to help the reader appreciate what the results are showing.

15. (small) It would be worthwhile to include tests for order effects.

·

Basic reporting

This paper presents the results of a study comparing four alternative visualization idioms for biological tree comparison. The study tries to focus on biological data and tasks, in contrast with other papers in the field which focuses more on general tasks.

The paper is well written, with clear and professional language.

The organization of the paper could be improved though, as the results are buried at the very end of the paper and should be highlighted right at the beginning in the abstract and introduction. Doing so could greatly improve the impact of the paper. Overall, the purpose of the paper is presented too late into the document, I'll suggest highlighting it right on the first paragraph of the introduction.

The journal format greatly affects the readability of these type of papers. Having tables and figures at the end is very detrimental for understanding a visualization paper. The paper shouldn't be published on this format (clarifying that this is the journal's fault and not the author's ). Having said that, the paper didn't include any figure illustrating the visualizations created, the interface, the type of questions, how the visualizations represented the chosen data, etc. This makes it very difficult to judge the results. The main visualization conferences even encourage a video and online demo so that the results are more reproducible, I strongly suggest these to be added.

The paper does a good job reporting the state of the art. It provides good comparisons with previous works, and leverages previous works by others to strengthen the results. However the choice to use the "ten main curation tasks" from a self referenced work is worrisome, this should be acknowledge as a treat to validity and justified in the paper.

For the related work and problem definition, I would also suggest clarifying the type of trees that are used on the study. Not all visualization techniques can be used with all types of trees described on the related work. The trees used for this study seems to be non aggregated trees, with a fixed hierarchy, without numeric attributes, and only labels on the leaves. This categorization is described on my work (already cited) Guerra-Gomez et al. 2012

Experimental design

The experiment motivation is sound, it is well justified and convinced me of the importance of the study. The research question is also well defined, and the methods applied make sense overall. The paper also includes a very good section acknowledging many of the limitations of the study and its treats of validity. The biggest of which was in my opinion the very small number of nodes involved in the tasks (less than 120). This contradicts the purpose of the experiment of finding the best type of visualization for biocomp tasks. Are bio scientists used to analyzed such tiny trees? if so this needs to be explained in the experiment design and more importantly this limitation needs to be mentioned when reporting the results. This is very important because the matrix visualization was viable because of the size of the dataset, and the results suggest that the matrix was the most effective idiom, therefore it should be clarified that this only applies for very small trees as the ones used. On the other hand, I applaud the decision of choosing an unfamiliar taxonomy for the tasks.

The paper doesn't describe how were the participants recruited, or if they were compensated for their participation.

It is also important to notice that randomization of the tasks, with such small number of participants won't be enough to counter learning effects.

It is very remarkable that participants have such broad domain experience, this adds value to the experiment. However, I would like to know what types of visualizations they have experience with, which could help to explain the results.

Many of the decisions made with the implementation were unfounded in my opinion. Examples of this are the choice of color hues and properties which could perceptually favor one task over the other. Not having any way of looking at the interface only make things worse

Validity of the findings

The results, despite being somewhat constrained in a way limited, are important, novel and impactful for the communities. The statistical methods seemed valid as well, although lots of discussions have being rising lately about the dangers of using p-values. Using confidence intervals could be a good addition to the paper, although I will leave this up to the authors and the journal.

The conclusions are well stated, but are very difficult to understand and process. They should be better summarized and presented in a more conclusive way at the very begining of the paper, if not in the abstract itself.

As I mentioned before, having the tables at the end of the paper is extremely detrimental for the reading experience. Moreover the tables are very difficult to interpret, there aren't any legends and they could be great improved by highlighting which differences were significant and which were the best alternatives for each task. Legends should help the reader by explaining if higher values are better or worse. The paper needs figures illustrating the 4 visualization alternatives, the main interface, and examples of how the chosen dataset + tasks were visualized. A video would be even better.

Neither the data, nor the prototype was shared, which greatly affects reproducibility. Many decisions made with the prototype seemed unjustified to me, and should be better defended, such as the choice of colors. Moreover, the paper claims to have left many interactions out of the prototype because "...the software... was not intended to be a final product". I strongly disagree with such statement, even the visualization mantra states that "zoom and filter" are critical for these type of applications, they aren't just bells and whistles that vendors can choose to add or not. Not having these interactions greatly change the scope of the results. This needs to be addressed in the paper

Additional comments

Good paper overall. I think it presents a valuable (yet constrained) results that is relevant for the community. The paper requires in my opinion major revisions that would make it significantly better, but these can be covered in another review cycle.

---

## Round 0.2 · Minor Revisions

Reviewer 1 has suggested some very minor revisions that should only take a few minutes to complete. Please make these revisions.

Reviewer 1 ·

Basic reporting

Since my first review, the authors have answered most of my concerns in the basic reporting section. However, while the figure has been split into four separate larger figures with improved legibility, they still have compression artefacts in them - this is probably just a battle to be had with the screen capture and pdf generating software but it does make the detail within them murky, especially in the matrix figure.

Experimental design

In their revision, the authors have tackled the comments in my original review as well as can be expected within the text. The reasoning for their method of iterating the tests by question rather than by visualisation is now clear. The acknowledged weaknesses in some of the visualisations' designs are now discussed at greater depth in the 'threat to validity' section, though 1) it would be good to state why the binding of datasets to methods seemed an acceptable risk, (or re-word it to say "trade-off" rather than risk, which then ties into the remainder of the sentence more clearly), and 2) state that learning effects are a risk to all small-scale studies, not just this one in particular.

The million dollar question is whether clearly stating the weaknesses is enough in everyone's view to compensate for them being there in the first place. My own opinion is that at the very worst, there may be a case to remove the agglomeration findings compared to the other visualisation types, as this was the visualisation type with the difficulty with colour choices.

Validity of the findings

I didn't have much to comment on that needed fixed or expanded on in this section. They've explained why the matrix method wasn't mentioned in some of the results discussion.

Additional comments

I'd like to thank the authors for taking the trouble to go through mine and the other reviewer's comments in such detail. I'd still be interested in more details on the artificial taxonomy generator, but maybe that's for another paper (people have made papers out of synthetic data generators).

One other point that the author's revision has brought up is that there have been quite a few tools for taxonomic comparison developed in the past, but they have mostly disappeared due to software/hardware obsolescence (some used unsecured java webstart, some only run on old operating systems, some just disappeared etc). How do they plan on stopping the same happening to their tool?

In summary, I would recommend a further minor revision.

·

Basic reporting

I'm satisfied with the actions taken by the authors to address the reviewers concerns. I'm still very worried though about the tables/figures at the end format, it shouldn't be used for this type of paper and the journal should do something about it.

Experimental design

I'm satisfied with the actions taken by the authors to address the reviewers concerns.

Validity of the findings

I'm still very worried though about the tables/figures at the end format, it shouldn't be used for this type of paper and the journal should do something about it.

---

## Round 0.3 · accepted · Accept

Thanks for addressing the reviewers' comments.